# Contribution of Capillary Zone Electrophoresis Hyphenated with Drift Tube Ion Mobility Mass Spectrometry as a Complementary Tool to Microfluidic Reversed Phase Liquid Chromatography for Antigen Discovery

**DOI:** 10.3390/ijms232113350

**Published:** 2022-11-01

**Authors:** Marie-Jia Gou, Murat Cem Kose, Jacques Crommen, Cindy Nix, Gael Cobraiville, Jo Caers, Marianne Fillet

**Affiliations:** 1Laboratory for the Analysis of Medicines (LAM), Department of Pharmacy, CIRM, University of Liege, Avenue Hippocrate 15, B36 Tour 4 +3, 4000 Liège, Belgium; 2Laboratory of Hematology, GIGA I3, University of Liège, Avenue de l’Hopital 11, B34, 4000 Liege, Belgium; 3Department of Hematology, Centre Hospitalier Universitaire (CHU) de Liège, Avenue de l’Hopital 1, 4000 Liege, Belgium

**Keywords:** capillary zone electrophoresis mass spectrometry, untargeted proteomics, drift tube ion mobility spectrometry, orthogonality

## Abstract

The discovery of new antigens specific to multiple myeloma that could be targeted by novel immunotherapeutic approaches is currently of great interest. To this end, it is important to increase the number of proteins identified in the sample by combining different separation strategies. A capillary zone electrophoresis (CZE) method, coupled with drift tube ion mobility (DTIMS) and quadrupole time-of-flight mass spectrometry (QTOF), was developed for antigen discovery using the human myeloma cell line LP-1. This method was first optimized to obtain a maximum number of identifications. Then, its performance in terms of uniqueness of identifications was compared to data acquired by a microfluidic reverse phase liquid chromatography (RPLC) method. The orthogonality of these two approaches and the physicochemical properties of the entities identified by CZE and RPLC were evaluated. In addition, the contribution of DTIMS to CZE was investigated in terms of orthogonality as well as the ability to provide unique information. In conclusion, we believe that the combination of CZE-DTIMS-QTOF and microfluidic RPLC provides unique information in the context of antigen discovery.

## 1. Introduction

Proteomics involves the large-scale study of all proteins expressed in a cell or living system under a particular environment at a given time [1,2]. Because of its dynamics, there might be quantitative and qualitative differences in protein expression in physiological versus pathological states. Therefore, one of the aims of clinical proteomic studies is to reveal proteins associated with a particular disorder in order to improve disease management. Moreover, the identification of those specific proteins could open new perspectives in the therapeutic care of patients, including cancers, by introducing, among others, new potential drug targets [2,3].

A mass spectrometry (MS)-based approach to the study of the proteome is relevant as it combines high selectivity, sensitivity, and throughput to unravel qualitative and quantitative information contained within inherently complex samples [4,5]. Moreover, by applying a bottom-up strategy for proteomics, sample complexity notably rises with the generation of a large number of tryptic peptides from proteins. Therefore, the addition of front-end separation techniques is necessary to achieve a better output in terms of ionization efficiency and detection sensitivity [5]. Among them, liquid chromatography (LC) coupled with MS is still the most widely used in biological applications, including proteomics, owing to its high separation efficiency and robustness. Furthermore, the introduction of miniaturized systems for LC separations, such as nanoflow LC-MS, has led to drastic improvements in terms of ionization efficiency and sensitivity [5,6,7]. 

As our goal is to maximize the number of identified entities, the combined use of different separation strategies could improve the overall identification rate, thus leading to more comprehensive proteome profiling [8,9,10,11]. For this purpose, capillary electrophoresis (CE), more specifically capillary zone electrophoresis (CZE), could bring additional insight into the analysis of complex proteome samples. The strengths of this technique lie in its ability to achieve a very efficient separation based on a different mechanism compared with LC, as well as its high ionization efficiency due to the nanoscale flow rate of CE effluent [8,9,10]. Several studies have already reported the usefulness of implementing multidimensional separation platforms with (nano)reversed-phase liquid chromatography (RPLC) and CE for the analysis of complex proteomes to obtain higher proteome coverage [8,9,10]. Indeed, despite the lower number of detected entities compared with LC, CE could still provide unique information which is of high interest in the context of comprehensive proteome profiling. 

In recent years, the use of ion mobility spectrometry (IMS) coupled with MS to study proteomes has become more and more widespread. There are many types of IMS instruments based on different principles, namely drift tube ion mobility spectrometry (DTIMS), high field asymmetric waveform ion mobility spectrometry (FAIMS), traveling wave ion mobility spectrometry (TWIMS), and trapped ion mobility spectrometry (TIMS). Their characteristics, applications, and comparisons are described in recently published reviews [11,12,13]. In this study, the focus was set on DTIMS coupled with quadrupole time-of-flight (QTOF) MS. The separation of ions in a DTIMS device is driven by their movement in a tube filled with an inert gas subjected to a low electric field. The ions could be differentiated depending on their charge, size, and shape [12,13]. Although the IM separation mechanism is similar to that of CZE, the potential orthogonality of the two systems has not been thoroughly studied yet [14]. Furthermore, the combination of DTIMS with CZE has lately proved to be beneficial in the context of targeted glycomics [15] and metabolomics [16] studies. 

To our knowledge, only a few studies concerning the application of CZE–DTIMS in the context of proteomics to improve separation efficiency are available. Indeed, our group previously reported that the use of data-independent acquisition (DIA) driven by DTIMS coupled with upstream CZE separation allowed the identification of unique peptides and proteins compared with the classical data-dependent acquisition (DDA) mode that did not involve the use of DTIMS [17]. Recently, Mast et al. developed a CZE-TIMS-MS workflow to characterize the stereochemistry of peptides in single cells by separating peptides containing D-amino acids at different positions in the sequence, which was not possible with CZE alone [18]. Those works prove that the added performance resulting from the coupling of IMS to CZE provides a significant contribution that could not be achieved by using CZE–MS alone. 

In this study, a CZE-DTIMS-QTOF method was developed and applied to the analysis of the surface proteome of a human multiple myeloma (MM) cell line, namely LP-1. Indeed, the final aim will be to apply this CZE-DTIMS-QTOF method in combination with our previously developed RPLC-Chip-DTIMS-QTOF method to the analysis of MM patient samples in order to identify new interesting antigens that could be targeted by innovative immunotherapeutic approaches (Appendix A). In this context, the enrichment of membrane proteins was performed by means of biotinylation. To identify as many proteins as possible, the use of a neutral coated capillary for CZE separation was compared with our previous method using a bare fused-silica (BFS) capillary. Additionally, the sample loading volume was also optimized to maximize protein identification. 

Next, the CZE-DTIMS-QTOF and RPLC-Chip-DTIMS-QTOF methods were compared in terms of the number of uniquely identified proteins, and further discussion about the nature of uniquely identified peptides and the orthogonality of the two systems is provided. 

Finally, the contribution of DTIMS in hyphenation with CZE was investigated as well as the ability to resolve co-migrating and co-isolated peptides. 

## 2. Results and Discussion 

### 2.1. Optimization of CZE-MS/MS Method 

An important drawback of CE is the limited loading capacity due to the low internal diameter of CE separation capillaries. Indeed, only a few nanoliters of samples are injected, which accounts only for 1–5% of the total capillary volume. To maximize sample loading, a preconcentration method must be implemented. With this aim in view, a dynamic pH junction method was developed in a previous study that allowed a 100-fold increase in detection sensitivity [17]. Subsequently, this method was applied in this study and further improved. 

#### 2.1.1. Use of Neutral Coated Capillaries

Because of the complexity of samples from cell proteome, there is interest in improving separation efficiency to maximize the identification rate. For this purpose, the use of neutral coated capillaries was considered. Indeed, coated capillaries tend to prevent peptide adsorption and interaction with the negatively charged BFS capillary wall, providing sharper peaks and thus improving peak capacity. Moreover, this could slow down the separation, leading to a larger migration window.

Triplicate injections of *E. coli* proteome digest at four different concentrations (from 0.3 µg/µL to 2.8 µg/µL) were performed using both BFS and polyvinyl alcohol (PVA)-coated capillaries. For example, triplicate injections of 0.25 µg of *E. coli* digest in the capillary resulted in the identification of 302 ± 8 peptides and 104 ± 6 proteins using the PVA capillary compared with 97 ± 6 peptides and 33 ± 3 proteins using the BFS capillary (Appendix A). Interestingly, the number of identified peptides and proteins was approximately multiplied by three with the use of PVA-coated capillaries. It is also worth noting that analyses performed using coated capillaries lead to the increase in migration window (Appendix A), which can partly explain the increase in the number of identifications.

Apart from the identification rate, a gain in terms of peak intensity and efficiency using PVA capillaries was also assessed (Figure 1). For this purpose, 25 peptides with various lengths and isoelectric points (pI) were randomly selected across the migration window (Appendix A). For a large majority of them, better performances were observed with the PVA-coated capillary.

Furthermore, peptides from the same dataset were employed to evaluate the peak capacity. So far, peak capacities measured by CZE have been reported on the basis of the measurement of full width at half maximum height (FWHM) [8,19,20]. However, to avoid any overestimation of peak capacity, peak width was taken at 13.4% of the maximum height. For this purpose, triplicate values of peak capacity were calculated based on injections performed using BFS and PVA capillaries (Appendix A). Interestingly, peak capacity was nearly doubled with the coated capillary (57.9 ± 0.9 for coated and 35.4 ± 6.6 for noncoated capillary) and was found to be much less variable (RSD = 1.6%) than the values obtained with the noncoated capillary (RSD = 18.2%). 

In view of the performance of PVA-coated capillaries, they were chosen for further studies on proteomic samples.

#### 2.1.2. Sample Injection Volume 

In a previous study using *E. coli* digest as a sample, 14% of the total capillary volume was filled with the sample during injection [17]. In the present study, different injection volumes were investigated with the aim of increasing the number of identified peptides and proteins without any negative impact on separation. 

Figure 2a shows that the injection of 20% of the total capillary volume provided the highest number of identified peptides using four different concentrations of *E. coli* digest (from 0.3 µg/µL to 2.8 µg/µL). The same tendency was also observed for proteins (Appendix A). As shown in the graph, increasing the injected sample volume up to 28% of the total capillary volume was not beneficial in terms of the number of identifications and repeatability. This decrease in the identification number despite the higher quantity of injected samples could be explained by the impact of the injection volume on the separation window. Indeed, Figure 2b shows that the migration window of the analyses dropped with the increase in the injected sample volume even though the signal intensity was proportional to the sample volume, especially from 20% to 28% of the total capillary volume. Consequently, a poorer separation leads to a lower number of identifications. In conclusion, the injected sample volume was set at 20% of the total capillary volume for further analysis using cell line samples.

### 2.2. Analysis of MM Cell Line Surface Proteins Using CZE-DTIMS-QTOF

#### 2.2.1. Uniqueness of Identifications between CZE-DTIMS-QTOF and RPLC-Chip-DTIMS-QTOF Methods

Three independent digests of LP-1 cell lines obtained after cell surface protein biotinylation and isolation were injected using the developed CZE-DTIMS-QTOF method. Acquisitions were performed using two different modes, namely DDA and DIA, to enhance protein identification [17,21]. To assess whether the use of CZE could be complementary to RPLC-Chip in terms of coverage, the obtained data using CZE-DTIMS-QTOF were compared to the analysis of LP-1 cell samples using RPLC-Chip. 

As an example, the overlaps of proteins between RPLC-Chip and CZE are displayed in Figure 3a. The number of identified entities was higher using RPLC-Chip, probably due to the high sensitivity provided by this microfluidic LC system associated with a robust spraying device. Interestingly, only 223 out of 3217 proteins were shared by the two techniques, which highlights their complementarity through different separation mechanisms, thus, different selectivities. In other words, 67% of the proteins that were identified by CZE-DTIMS-QTOF were unique to this method. The same tendency was observed with peptides (Appendix A). Among all the proteins that were identified in LP-1 cell lines, approximately 14% were uniquely contributed by CZE-DTIMS-QTOF, as displayed in Figure 3b. By analyzing the three samples of LP-1 cells, an average percentage of 13.3% ± 0.6% (%RSD: 4.9) was obtained for all proteins identified by CZE. Those results proved that the use of CZE could bring additional information in terms of identified proteins and peptides that could not be covered by RPLC-Chip only, despite its high sensitivity. 

Therefore, the interest in CZE-DTIMS-QTOF as a complementary tool to enhance proteome coverage in combination with RPLC-Chip was demonstrated. 

#### 2.2.2. Orthogonality of RPLC-Chip and CZE

Previous results showed the ability of CZE-DTIMS-QTOF to provide complementary information to RPLC-Chip in terms of the number of identified peptides and proteins. Furthermore, in order to determine to what extent the use of both techniques could bring complementary information, the orthogonality factor between RPLC-Chip and CZE was evaluated. Indeed, a high degree of orthogonality for two separation systems corresponds to two uncorrelated selectivities. 

The orthogonality of two-dimensional (2D) separations, especially 2D-LC, is well described in the literature with different methods to calculate the orthogonality factor [22,23,24,25]. As a matter of fact, none of the orthogonality metrics appears to be the gold standard method, and each has its own strengths and weaknesses [26]. Instead of relying on one of these metrics, their combination might provide a better evaluation [24,26]. Therefore, in this study, orthogonality was first evaluated by a widely used approach, namely the geometric surface coverage (SCG) method [24,25]. Moreover, another set of orthogonality metrics described by Ruotolo et al. [27] was also employed to confirm or infirm the tendency observed with the first method. 

The SCG approach is a method used to characterize separation orthogonality described by Gilar et al. [25]. In this approach, the occupation of 2D separation space by the identified peptides is estimated, which reflects the orthogonality of both systems. Indeed, a higher percentage of coverage leads to an increase in orthogonality. Gilar and al. [25] stated that a 10% surface coverage represents a nonorthogonal system. Since 100% coverage is hypothetically impossible to achieve in the context of separations, they determined that a surface coverage of 63% is considered optimal. 

This method has proved to be intuitive, easy to perform, and robust. However, it is worth noting that a higher variability of the orthogonality factor might be observed when the sample size is limited. Therefore, orthogonality calculations were performed using three independent samples, and a number of peptides were pooled to obtain a maximum of data points having a pair of retention and migration times.

A second method used in this study to calculate the orthogonality factor was also based on the separation space occupied by identified peptides. This method was described by Ruotolo et al. [27] and was originally used to calculate the orthogonality between DTIMS and MS. 

In this study, 223 pairs of retention and migration times were employed to calculate the orthogonality between CZE and RPLC-Chip. Figure 4a shows the application of method 1 using our CZE × RPLC-Chip data. The separation space containing the 223 peptides was divided into a 15 × 15 matrix, and a surface coverage of 48% was obtained. Subsequently, an orthogonality factor was calculated using equation (2), and a value of 0.66 was obtained. 

Figure 4b displays the use of method 2 to determine the orthogonality factor between RPLC-Chip and CZE. The peptides were included in a space between −31.1% and +35.8% around the linear regression line. Therefore, 66.9% of the separation space was occupied by peptides providing an orthogonality factor of 0.67.

Interestingly, the orthogonality factors calculated with both methods were consistent, which demonstrates the reliability of the results. 

Thus, high orthogonality between RPLC-Chip and CZE was obtained, as reflected in the values of the orthogonality factor. Indeed, separation in RPLC-Chip is based on analyte partition between stationary and mobile phases depending on its hydrophobicity. In contrast, separation in CZE relies on analyte mobility in an electrolyte based on its charge-to-size ratio under the influence of an electric field. Due to the difference between these separation mechanisms, orthogonality was expected to be important.

#### 2.2.3. Physicochemical properties of identified peptides in RPLC-Chip and CZE

In addition to the complementarity of RPLC and CZE in terms of the number of identifications as well as the high orthogonality factor, the difference in nature of identified peptides using both RPLC-Chip-DTIMS-QTOF and CZE-DTIMS-QTOF was investigated. 

For this purpose, several physicochemical properties were chosen for this comparison, including peptide length, peptide mass, precursor ion m/z ratio, grand average of hydropathy (GRAVY) score, mass/length ratio, and pI. The difference between RPLC-Chip and CZE in terms of the nature of identified peptides, as well as the level of contribution of each variable, was evaluated through a multivariate approach, namely principal component analysis (PCA). As shown in Figure 5b, the combination of components 1 and 2 could explain 72.3% of the variability between RPLC-Chip and CZE. Indeed, component 1 described 51% of the variability, which gathers peptide mass, peptide length, and precursor ion m/z ratio as main variables. In addition, the GRAVY score was the main variable included in component 2, which accounted for 21.3% of the variability. 

Figure 5a displays the PCA density plot evaluating the differences between the peptides identified in RPLC-Chip and CZE with regard to their physicochemical properties. Concerning the peptides identified in RPLC-Chip, most of them seem to be described by variables from PC1, including peptide length, mass, and precursor ion. As a result, the peptides identified in RPLC-Chip were longer and heavier. On the other hand, the peptides identified using CZE are more correlated to PC2, which corresponds to the GRAVY score as well as pI. The latter is negatively correlated to PC1, according to Figure 5b. Consequently, most peptides identified by CZE tend to be more polar and basic. Following those observations, a univariate analysis comparing the above variables in RPLC-Chip and CZE was carried out (Appendix A), and a similar conclusion to multivariate analysis was revealed. Moreover, those observations were in accordance with other studies [10,28]. 

Interestingly, the complementarity between RPLC-Chip and CZE could be explained partly by the physicochemical properties that were described in this study. The distinct selectivities provided by those two techniques surely contributed to those differences in terms of characteristics. Those differences could be explained since peptides with higher masses usually interact more strongly with RPLC stationary phase leading to more efficient separation, unlike low molecular weight peptides, which more often co-elute. On the other hand, CZE is better suited to the separation of more polar entities. Additionally, basic peptides tend to be positively charged in the acidic background electrolyte (BGE) used in our system and are more likely to be separated and fully ionized in the source.

### 2.3. Contribution of DTIMS to CZE Analysis in Proteomics 

#### 2.3.1. Orthogonality CZE x DTIMS

Our previous results proved that the use of DIA driven by DTIMS could provide additional unique information in terms of the number of identified entities compared with CZE [17]. To further assess the benefit provided by DTIMS separation, the orthogonality between DTIMS and CZE was evaluated in this study using the two methods described in Section 2.2.2.

In this comparison, 2158 pairs of migration times and drift times were used to determine the orthogonality between CZE and DTIMS. First, method 1 was applied, and a 47 × 47 matrix was obtained for the separation space of the 2158 peptides (Figure 6a). After bin counting, a surface coverage of 31.3% was determined, which was converted to an orthogonality factor of 0.46 via equation (2). Regarding method 2, migration/drift time pairs were plotted, followed by linear regression analysis. The separation space around the resulting fit-line was found to range from −17.9% to +22.1%. Consequently, the obtained orthogonality factor was 0.40.

As in the case of RPLC-Chip × CZE, the calculated values of orthogonality between CZE and DTIMS could be stated as reliable since the result using method 1 was in accordance with that obtained with method 2. These values of orthogonality indicate that CZE × DTIMS is less orthogonal than RPLC × CZE but more orthogonal than DTIMS × MS, which has an orthogonality factor of 0.33 using method 1. As stated earlier, CZE and DTIMS possess similar characteristics in their separation mechanisms, including charge and size dependency, as well as the application of an electric field to drive the separation. The main distinction is the separation medium which consists of a liquid electrolyte for CZE and an inert gas for DTIMS [14]. Interestingly, an orthogonality of 0.40-0.46 was observed, which means that CZE and DTIMS selectivities are different despite having close separation principles. This observation suggests that hyphenating CZE and DTIMS theoretically makes sense. It is worth noting that the DIA method driven by the use of DTIMS does not require any abundance threshold for precursor fragmentation, which could explain the increase in the number of identifications in our CZE-DTIMS-QTOF method compared with the use of the classical DDA mode. Nevertheless, even though the higher number of identifications in CZE-DTIMS-QTOF could be related to the use of DIA, the DTIMS separation definitely plays a role because of its unique selectivity. Therefore, a better comprehensive proteome profiling could be obtained by using CZE-DTIMS-QTOF instead of CZE-QTOF only.

#### 2.3.2. Co-migrating and Co-isolated Peptides

The use of separation systems prior to MS detection aims at reducing proteome sample complexity. However, even a high-resolution separation technique such as CZE is not sufficient since many co-migrations still occur in the CZE dimension. This overlapping of analytes could lead to ion suppression. Furthermore, in the MS dimension, the presence of isobaric species could lead to their co-isolation in the quadrupole and then undergo co-fragmentation. Consequently, the acquisition of chimeric MS/MS spectra could be observed, which negatively impacts further protein identification [29].

In this study, the ability of DTIMS to resolve co-migrating and co-isolated peptides was evaluated. For this purpose, peptides of interest were searched among the identified peptides by analyzing digested proteins extracted from the LP-1 cell line. It is worth noting that peptides were considered co-migrating if their difference in migration time was not more than 0.1 min. In the same way, peptides were regarded as co-isolated if their m/z ratio difference was not more than 0.001 m/z.

For instance, three peptides with the following sequences and charges were found to be co-migrating and co-isolated within our analysis (Figure 7): LGKIIINNKnFDK (2+), YEImDGAPVKGESIPIRLFLAGYDPTPTMR (4+), and IAFWLQWFnSFVNPLLYPLCHKRFQK (4+). All three peptides migrated at 38.2 min, and their m/z ratio was 839.193 m/z. Even though CZE and MS dimensions were not sufficient to resolve those three species, the use of DTIMS allowed their complete separation with drift times of 25.3, 27.5, and 28.6 ms, respectively. Moreover, those three peptides belonged to three different proteins. Therefore, in this case, the use of CZE-DTIMS-QTOF allowed the identification of three different proteins instead of one if only CE-MS were used. 

In addition, as an example, 53 pairs of co-migrating and co-isolated peptides were detected among the total number of peptides identified in one CZE-DTIMS-QTOF analysis. Then, 50 pairs out of 53 were resolved in the DTIMS dimension by means of the additional separation space provided by DTIMS. Therefore, the interest in using CZE-DTIMS-QTOF to enhance proteome coverage is high since this system could separate co-migrating and co-isolated peptides that would not be resolved without the contribution of DTIMS.

## 3. Materials and Methods

### 3.1. Chemicals and Reagents 

ULC-MS grade methanol (MeOH), water, formic acid (FA), and acetic acid (HAc) were purchased from Biosolve (Valkenswaard, the Netherlands). Ammonia solution (25% *w/v*, NH_4_OH) and fetal bovine serum (FBS) were obtained from Merck (Darmstadt, Germany). Dulbecco’s Modified Eagle Medium (DMEM) and Penicillin/Streptomycin stock, 10,000/10,000 used for cell culture, were from Lonza (Verviers, Belgium). 

Precut BFS and PVA-coated capillaries were purchased from Agilent Technologies (Waldbronn, Germany). Standard *E. coli* proteome digest was obtained from Waters (Dublin, Ireland), and LP-1 cells were kindly provided by H. Jernberg-Wiklund (Uppsala University, Uppsala, Sweden).

### 3.2. Cell Culture 

LP-1 cells were grown in DMEM culture medium supplemented with 10% (*v/v*) FBS and 1% (*v/v*) Penicillin/Streptomycin 10,000/10,000 solution. Cultures were kept in an incubator set at 37°C with the addition of 5% carbon dioxide. Cell density and viability were determined using trypan blue staining in a 1:1 dilution. The homogenized cell/trypan blue solution was then loaded onto a hemocytometer, and automatic counting was performed by means of a Corning cell counter (CytoSMART, Eindhoven, The Netherlands). Additionally, LP-1 cells were tested mycoplasma-free.

### 3.3. Sample Preparation 

Medium containing floating LP-1 cells was centrifuged at 500 x g to collect cell pellets. Three replicates of ten million LP-1 cells were harvested, and biotinylation was directly performed on live cells. After biotinylation, cells were lysed, and biotin-labeled proteins were isolated using NeutrAvidin agarose beads (Thermo Fischer, Waltham, MA, USA).

Protein elution from beads, digestion, and peptide cleanup prior to MS analysis were performed using several buffers from iST kits (PreOmics, Martinsried, Germany). Afterward, the cleaned peptide digests were evaporated at 45°C until dryness, and samples were kept at −80°C. Prior to injection, samples were resuspended in 20 µL of 200 mM ammonium acetate (NH_4_Ac) at pH 9 to perform preconcentration [17].

### 3.4. Instrumentation and Working Conditions

#### 3.4.1. Instrumentation 

All CZE experiments were performed using a G7100 CE system (Agilent Technologies, Waldbronn, Germany) hyphenated with a 6560-hybrid DTIMS-QTOF (Agilent Technologies, Waldbronn, Germany) with detection in positive ionization mode for all acquisitions. 

Hyphenation of both instruments was achieved by the use of a coaxial sheath liquid interface sprayer (Agilent Technologies). The sheath liquid consisted of MeOH/H_2_O/FA (80:20:0.5, *v/v/v*) and was pumped into the sprayer at a flow rate of 0.38 μL/min to guarantee stable spraying. The working conditions of RPLC-Chip are described in [7].

#### 3.4.2. Optimized CZE-DTIMS-QTOF Method for Cell Sample Analysis

For CZE separation, a 50 µm ID PVA-coated capillary (Agilent Technologies, Waldbronn, Germany) with a length of 125 cm was employed for cell sample analysis. Before each experiment, the capillary was first flushed using deionized water for 10 min followed by a 10 min flush using BGE, which consisted of 2.5% (*v/v*) HAc. The separation voltage was set at + 30 kV, and an inlet pressure of −5 mbar was applied during the run. Samples were hydrodynamically introduced by applying a pressure of 100 mbar for 360 s. 

To ensure correct ionization, the capillary voltage in the electrospray source was set at 4 kV. The pressure of the nebulizing gas and the flow rate of the drying gas were set at 6 psi and 4 L/min, respectively. The gas temperature was set at 300 °C. 

Both CE and MS instruments were controlled using MassHunter LC/MS Data Acquisition software B.09.00 (Agilent Technologies). 

DDA, as well as DIA modes, were used in this study. All acquisition parameters are described in [17] except for the number of precursors selected in DDA, which was 15.

### 3.5. Data Analysis 

Extracted ion chromatograms were obtained using MassHunter Qualitative Analysis Navigator B.10.00 (Agilent Technologies). Information, including peak intensity and FWHM for each peak, was also obtained from the same software after manual peak integration. 

Peak capacity was calculated using the calculation approach described by Nys et al. [7]. Briefly, FWHM values were first converted into peak widths at 13.4% (W4σ). Then, the following equation (1) was applied to calculate peak capacity:(1)$$Peak\;capacity=1+\frac{MT_{L}-MT_{F}}{\frac{\sum^{\;}W_{4\sigma }}{n}}$$
where *MT_F_* and *MT_L_* are the migration times of the first peptide and the last peptide of the separation window, while n is the total number of peptides through the separation window. 

Regarding physicochemical properties, peptide masses, pIs, and precursor ions were obtained from experimental data. Peptide lengths, mass/length ratios, and GRAVY scores were calculated using MS Excel. 

For graphical representations, electropherograms were generated using OriginPro 2017 B9.4.0.220. All other plots were obtained using GraphPad Prism 8.0.1 (La Jolla, CA, USA). Linear regression and PCA were performed using JMP Pro 15.2.0. Density plots, as well as calculation and representation of orthogonality involving bin counting, were performed using Python 3.10. All statistical analyses were performed using GraphPad Prism 8.0.1. Mann–Whitney test was applied for comparisons, and differences with a *p* value not higher than 0.05 were considered significant.

### 3.6. Protein Identification 

To generate peptide/protein identification lists, DIA MS/MS files were first reprocessed using IM-reprocessor to recalibrate the mass axis using 121.0509 and 921.0098 as reference masses. Afterward, CZE and IMS dimensions were smoothed using PNNL PreProcessor software (Pacific Northwest National Laboratory, Richland, WA, USA). Furthermore, a list of ion features was generated by the application of a 4D-ion mobility feature extraction algorithm through the use of IM Browser (Agilent). The selected isotope model was “peptides” with a charge state between 2–7 and ion intensity above 50. Finally, PKL files (compatible with Spectrum Mill) were generated, containing the extraction and alignment of MS/MS spectra with similar retention times (±10 s) and drift times (±0.5 ms).

DDA MS/MS files and exported pkl files from DIA MS/MS data files were imported into Spectrum Mill Software (Agilent Technologies) for peptide sequencing. Carbamidomethylation of cysteines was selected as fixed modification while oxidation of methionines, deamidation of asparagines and glutamines, as well as carbamidomethylation-propionylation of lysines (only for biotinylated samples) were selected as variable modifications. Trypsin (for *E. coli* digest) or LysineC/Trypsin (for MM cell line) was set as the digestion enzyme, and a maximum of 2 missed cleavages was allowed. Mass tolerance for precursor and product ions was set at 20 and 50 ppm, respectively. Identified peptides were considered a reliable hit when having a fragmentation score > 5 and spectrum purity index (SPI) > 50%.

### 3.7. Orthogonality Calculation 

The SCG method, described as “method 1” in the above sections, was used for orthogonality calculation [24,25]. First, retention times were plotted against migration times after normalization. Then, the whole separation space was divided into a number of squared bins determined by the number of identified peptides. All the bins containing at least one peptide were summed to obtain the number of occupied bins, which was subsequently divided by the total number of bins to assess the surface coverage and, thus, orthogonality. The following equation was used to calculate the orthogonality factor (2):
(2)$$Orthogonality\;factor=\frac{\sum^{\;}occupied\;bins-\sqrt{N}}{0.63\times N}$$
where *N* is the number of data points. 

A second method, described by Ruotolo et al. [27], was used as “method 2” to calculate the orthogonality factor. In this case, retention times were plotted against migration times without normalization. After performing linear regression, the maximum deviation of peptide migration times from the regression line was calculated, which determined the percentage of space occupied by the whole dataset and, thus, the orthogonality factor. Finally, 95% of outliers were eliminated to prevent any overestimation.

## 4. Conclusions

Throughout this study, the aim was to develop a CZE-DTIMS-QTOF method for the analysis of biotinylated MM cell digests to identify potentially interesting antigens to be targeted. In addition, the usefulness of CZE as a complement to RPLC-Chip for the identification of proteins and peptides was assessed, as well as the contribution of DTIMS in this separation system. 

First, a CZE-DTIMS-QTOF method was optimized at the CZE level, and the use of PVA-coated capillaries was implemented, as well as the injection of a sample volume at 20% of the total capillary volume, in order to maximize the number of identifications. 

Then, LP-1 cell lines were analyzed using the optimized method, and the results were compared with the data acquired using an RPLC-Chip method in terms of the uniqueness of identifications and their nature. Interestingly, the use of CZE allowed the identification of 13.3% of extra proteins that could not be covered by RPLC, which highlights the additional input of CZE. Additionally, the orthogonality between RPLC and CZE was also calculated and appeared to be high, as expected due to their distinct separation mechanisms. 

Finally, the contribution of DTIMS to CZE analysis was also assessed by the calculation of their orthogonality. The latter was lower than that between RPLC and CZE but higher than that between DTIMS and MS. Therefore, the combined use of DTIMS and CZE, despite their similar separation principles, makes sense. Moreover, the addition of DTIMS to CZE and QTOF allowed the separation of co-migrating peptides in the CZE dimension as well as co-isolated peptides in the MS dimension. 

In conclusion, our developed CZE-DTIMS-QTOF method could be applied in the context of MM antigen discovery because of its ability to provide unique information that could not be yielded by the sole use of traditional methods.

## Figures and Tables

**Figure 1 ijms-23-13350-f001:**
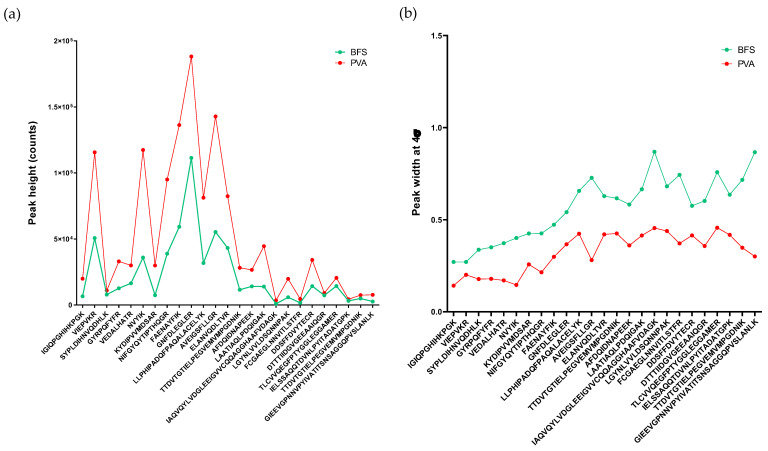
(**a**) Comparison of peak intensities and (**b**) peak widths at 4σ of 25 random peptides extracted from *E. coli* digest throughout the migration window and analyzed using both BFS capillary and neutral coated PVA capillary.

**Figure 2 ijms-23-13350-f002:**
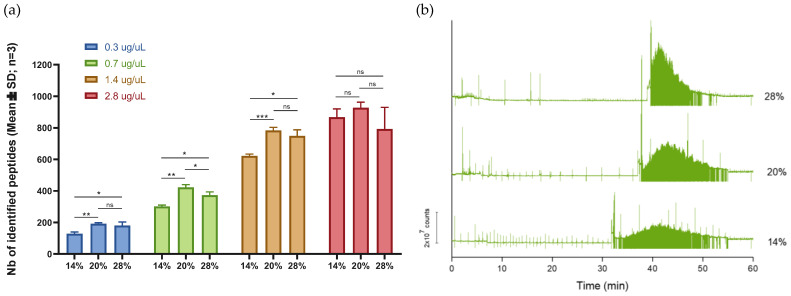
(**a**) Influence of the injected sample volume on the number of identified peptides using four different concentrations of *E. coli* digest. (Mann-Whitney test; ns, *p* > 0.05; *, *p* ≤ 0.05; **, *p* ≤ 0.01 and ***, *p* ≤ 0.001) (**b**) Total ion count illustrating the migration window of *E. coli* digest with injected sample volumes of 14%, 20%, and 28% of the total capillary volume.

**Figure 3 ijms-23-13350-f003:**
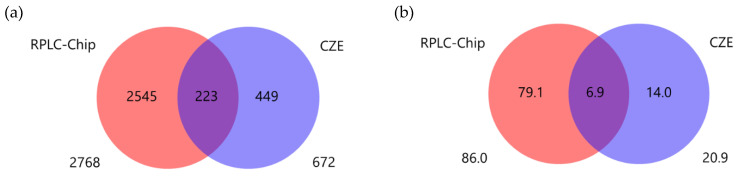
(**a**) Uniqueness of identified proteins and (**b**) percentage of identified proteins by analyzing LP-1 cell lines using RPLC-Chip and CZE coupled with DTIMS-QTOF.

**Figure 4 ijms-23-13350-f004:**
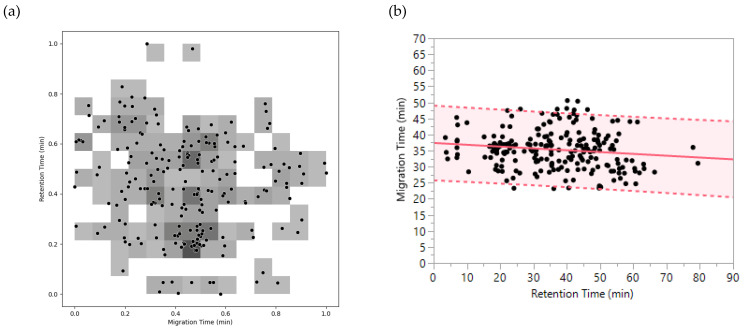
Orthogonality between RPLC-Chip and CZE using methods 1 and 2. (**a**) Method 1: Plot illustrating the coverage of the separation space by peptides through shaded bins. (**b**) Method 2: Plot illustrating the distribution of peptides around linear regression. The pink section indicates the separation space occupied by peptides.

**Figure 5 ijms-23-13350-f005:**
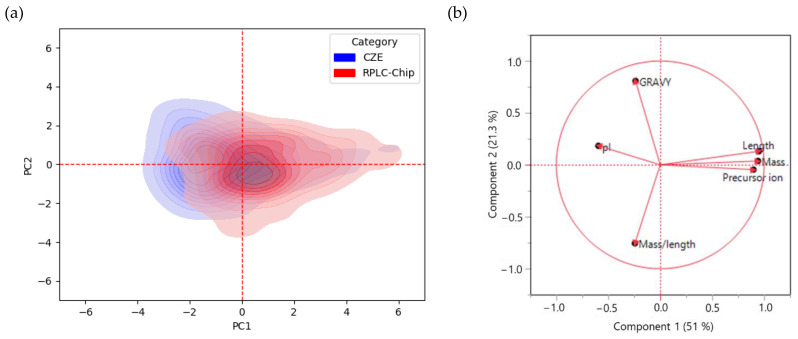
(**a**) Density PCA plot representing peptides identified in RPLC-Chip (Red) and CZE (Blue) according to their physicochemical properties. (**b**) Loading plot including the physicochemical properties as variables.

**Figure 6 ijms-23-13350-f006:**
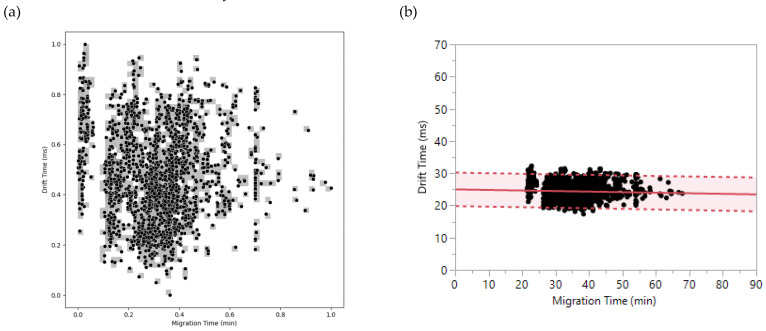
Orthogonality between CZE and DTIMS using methods 1 and 2. (**a**) Method 1: Plot illustrating the coverage of separation space by peptides through shaded bins. (**b**) Method 2: Plot illustrating the distribution of peptides around the regression line. The pink section indicates the separation space occupied by peptides.

**Figure 7 ijms-23-13350-f007:**
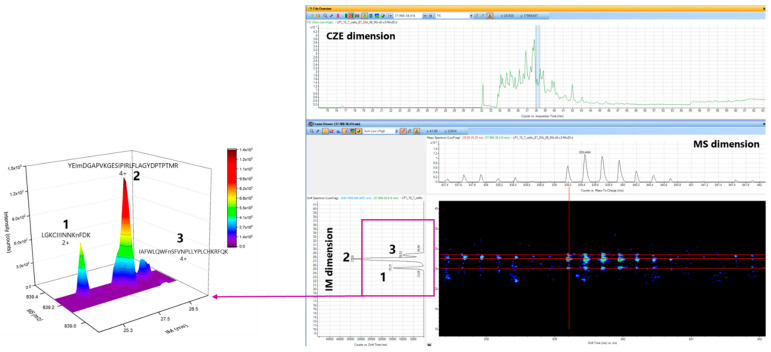
Example of three co-migrating and co-isolated peptides in CZE and MS dimensions. The separation of the three species was obtained in the IMS dimension allowing their identification.

## Data Availability

The data presented in this study are available on request from the corresponding author. The data are not publicly available due to their size.

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
