# Peer review of "Contribution of Capillary Zone Electrophoresis Hyphenated with Drift Tube Ion Mobility Mass Spectrometry as a Complementary Tool to Microfluidic Reversed Phase Liquid Chromatography for Antigen Discovery"

_ijms, 2022, doi:10.3390/ijms232113350_

Round 1

Reviewer 1 Report

The manuscript describes the advantages of hyphenation of capillary zone electrophoresis (CZE), drift tube ion mobility (DTIMS) and quadrupole time-of-flight mass spectrometry (QTOF). The authors evaluated the proposed strategy by analyzing E.Coli cells digest and then applied the optimized method for antigen discovery using the human myeloma cell line LP-1. Overall, the manuscript is well written and the results are sound. I recommend the authors to address the following comments before accept the manuscript for publication in IJMS.

Major comments:

1) The use of neutral coated capillaries is well known to prevent the interaction/adsorption of charged compounds to the capillary wall (which is usually negatively charged in bare fused silica capillaries). This strategy is widely used, especially for biological samples. To me, this is the main contributing factor to the increasing number of proteins and peptides observed when PVA coated capillaries are used instead of BFS ones. And not the increased separation window as stated by the authors.

2) for the data shown in Figure1B, why “Peak width at 13.4% of the maximum height” was used instead of the conventional 50% (FWHM, full width at half maximum) or 10% (that is widely used for a better representation when peak tailing is pronounced)?    

3) The results point to a significant variation of peptides and proteins identified with CZE-DTIMS-MS and with RPLC-Chip-DTIMS-MS. Although the separation mechanisms in CZE and RPLC-Chip are very different, and could support the finds, how can the author eliminate (or separate) the contribution/effect of the interface with DTIMS-MS (the transferring from the liquid phase separation technique to the gas phase separation/detection)? For example, ionization process, ion suppression, fragmentation, aggregation, and so on.

4) It would be helpful for the readers if the authors add schematic diagrams showing both setups used (CZE-DTIMS-MS and RPLC-Chip-DTIMS-MS) and the interfaces. If manuscript length is an issue, it can be added to the supplementary material.

5) For the data shown in Figure 6A, the authors used equation (1) to calculate orthogonality factor (as mentioned in line 315). However, equation one refers to peak capacity. I believe the authors used the correct equation (2) but have a typo in the text. Please correct it.

Minor comments:

- Caption of Figure S2: The term “total ion chromatograms” are probably used improperly. If it refers to the signal recorded for CE separations, the term “electropherograms” would be more appropriated. Another option it to use “total ion count” in case the authors prefer to refer to it as a MS result. 

- The same applies to the caption of Figure 2B.

Author Response

Dear Reviewer,

At first, we would like to thank you for their interesting comments. We totally agree with their remarks and we did our best to adapt the manuscript according to their suggestions. All corrections are displayed as tracked changes in the manuscript. Concerning the reviewers’ questions, they can be found in this document.

Reviewer 1

The manuscript describes the advantages of hyphenation of capillary zone electrophoresis (CZE), drift tube ion mobility (DTIMS) and quadrupole time-of-flight mass spectrometry (QTOF). The authors evaluated the proposed strategy by analyzing E.Coli cells digest and then applied the optimized method for antigen discovery using the human myeloma cell line LP-1. Overall, the manuscript is well written and the results are sound. I recommend the authors to address the following comments before accept the manuscript for publication in IJMS.

Major comments:

  • The use of neutral coated capillaries is well known to prevent the interaction/adsorption of charged compounds to the capillary wall (which is usually negatively charged in bare fused silica capillaries). This strategy is widely used, especially for biological samples. To me, this is the main contributing factor to the increasing number of proteins and peptides observed when PVA coated capillaries are used instead of BFS ones. And not the increased separation window as stated by the authors.

This comment was addressed in the manuscript by stating that the main contribution to the increase in the number of identified peptides is due to the prevention of peptide adsorption by the use of a neutral coated capillary.

  • for the data shown in Figure1B, why “Peak width at 13.4% of the maximum height” was used instead of the conventional 50% (FWHM, full width at half maximum) or 10% (that is widely used for a better representation when peak tailing is pronounced)?

In this work, peak capacity of BFS and PVA capillaries was calculated for their comparison. Since peak width at 13.4% of the maximum height was used to calculate peak capacities, we have kept those values to describe peak width in Figure 1B to stay in accordance within our study.

  • The results point to a significant variation of peptides and proteins identified with CZE-DTIMS-MS and with RPLC-Chip-DTIMS-MS. Although the separation mechanisms in CZE and RPLC-Chip are very different, and could support the finds, how can the author eliminate (or separate) the contribution/effect of the interface with DTIMS-MS (the transferring from the liquid phase separation technique to the gas phase separation/detection)? For example, ionization process, ion suppression, fragmentation, aggregation, and so on.

In this study, comparisons between RPLC-Chip and CZE were performed using the same instrument and method for downstream acquisition and detection (DTIMS-QTOF). Nano-electrospray ionization was employed for both setup but interfacing is certainly different due to the use of two distinct separation techniques (Chip Cube for RPLC-Chip and Triple tube sprayer for CZE). Besides, spraying flowrate in RPLC-Chip interface and CZE interface corresponds to 300 nL/min and 3800 nL/min, respectively. We agree with your comment stating that those distinct interfaces might influence our results. However, in order to perform our study, the use of those specific interfaces for each technique was mandatory.

In this study, the main objective was to increase the number of identified peptides in the context of antigen discovery by means of complementary techniques namely RPLC-Chip and CZE. Therefore, no evaluation on the possible influence of their respective interfaces on our results was performed.

  • It would be helpful for the readers if the authors add schematic diagrams showing both setups used (CZE-DTIMS-MS and RPLC-Chip-DTIMS-MS) and the interfaces. If manuscript length is an issue, it can be added to the supplementary material.

As suggested, a schematic diagram illustrating our CZE-DTIMS-QTOF and RPLC-Chip-DTIMS-QTOF setups was added to the supplementary data as Figure S1.

  • For the data shown in Figure 6A, the authors used equation (1) to calculate orthogonality factor (as mentioned in line 315). However, equation one refers to peak capacity. I believe the authors used the correct equation (2) but have a typo in the text. Please correct it.

The equation number was corrected from 1 to 2 in the manuscript.

Minor comments:

  • Caption of Figure S2: The term “total ion chromatograms” are probably used improperly. If it refers to the signal recorded for CE separations, the term “electropherograms” would be more appropriated. Another option it to use “total ion count” in case the authors prefer to refer to it as a MS result.
  • The same applies to the caption of Figure 2B.

“Total ion chromatogram” in the caption of Figure S2 and Figure 2B was replaced by “total ion count” as suggested.

Reviewer 2 Report

This manuscript highlights the added value of coupling capillary zone electrophoresis with drift tube ion mobility  and mass spectrometry in the frame of proteomics. The study is well explained and builds further on previous work of the same group.

The following minor comments can be made:

Page 2 line 82 (and also page 3 line 533 and page 12 line 433 and elsewhere throughout the paper): is this reference number 21 correct? It does not seem to refer to a publication on the group. Maybe reference 24 was meant to be used.

Page 3 lines 123-124: the numbers of peptides and proteins do not seem to correspond to the figures/bars shown in Figure S1 (blue bars).

In general, the reference numbers need to be checked since references 31 and 32 were used in the text, but do not feature in the reference list.

Author Response

Dear Reviewer,

At first, we would like to thank you for their interesting comments. We totally agree with their remarks and we did our best to adapt the manuscript according to their suggestions. All corrections are displayed as tracked changes in the manuscript. Concerning the reviewers’ questions, they can be found in this document.

Reviewer 2

This manuscript highlights the added value of coupling capillary zone electrophoresis with drift tube ion mobility  and mass spectrometry in the frame of proteomics. The study is well explained and builds further on previous work of the same group.

The following minor comments can be made:

  • Page 2 line 82 (and also page 3 line 533 and page 12 line 433 and elsewhere throughout the paper): is this reference number 21 correct? It does not seem to refer to a publication on the group. Maybe reference 24 was meant to be used.

All the references were checked and corrected

  • Page 3 lines 123-124: the numbers of peptides and proteins do not seem to correspond to the figures/bars shown in Figure S1 (blue bars).

The exact number of peptides and proteins mentioned in line 123-124 corresponds to the green bar in Figure S1 which is the E.coli sample at a concentration of 0.7 µg/µL.

  • In general, the reference numbers need to be checked since references 31 and 32 were used in the text, but do not feature in the reference list.

All references were checked carefully and corrected.